# Potential Effects of Resistant Exercise on Cognitive and Muscle Functions Mediated by Myokines in Sarcopenic Obese Mice

**DOI:** 10.3390/biomedicines10102529

**Published:** 2022-10-10

**Authors:** Gahyun Lim, Heaji Lee, Yunsook Lim

**Affiliations:** Department of Food and Nutrition, Kyung Hee University, Seoul 02447, Korea

**Keywords:** sarcopenic obesity, myokines, resistant exercise, cognitive function, muscle function

## Abstract

Recently, it has been demonstrated that in sarcopenic obesity (SO), physical activity could improve cognitive functions. Moreover, previous studies suggested that muscle contraction could influence cognitive function via myokines. This study investigated the potential effects of resistant exercise on cognitive and muscle functions in SO. SO was induced by a high-fat diet treatment for 8 weeks in 8-month-old male C57BL/6J mice. Then, resistant exercise (ladder climbing) for 8 weeks was performed. Muscle and cognitive function tests and morphological analysis were conducted. The protein levels of myokines were investigated in muscle, plasma, and the hippocampus in sarcopenic obese mice. Muscle and cognitive functions were significantly elevated in the obesity-exercise group (EX) compared to the obesity-control group (OB). Interestingly, muscle function was positively correlated with cognitive function. Abnormal morphological changes in the hippocampus were ameliorated in EX compared to OB, but not in the muscle. Protein levels of cognitive function-related myokines and energy metabolism-related markers in EX were significantly elevated in both muscle and hippocampus compared to those in OB. Interestingly, the protein level of brain-derived neurotrophic factor (BDNF) in EX was simultaneously increased in all tissues including muscle, plasma, and hippocampus compared to that in OB. In conclusion, modulation of muscle-derived cognitive function-related myokines in various pathological conditions via a resistant exercise could be a possible way of relieving muscle and cognitive dysfunction.

## 1. Introduction

The prevalence of sarcopenic obesity, which is the combination of obesity and sarcopenia, is increasing in the aged population. It is a predominant geriatric syndrome that accompanies detrimental complications from both obesity and sarcopenia [1]. The decline of both muscle and brain function is a representative complication of sarcopenic obesity. In particular, a recent cross-sectional study demonstrated that sarcopenic obesity can be a potential indicator of cognitive impairment [2]. Reduction in cognitive and muscle functions seriously hinders an individual’s independence and quality of life [3]. In this regard, investigating how to mitigate declined cognitive and muscle functions in sarcopenic obesity is important

A recent meta-analysis suggested that sarcopenia is associated with cognitive dysfunction [4]. The correlation between obesity and cognitive dysfunction has been shown in several investigations [5,6]. Exacerbated insulin resistance [7], energy metabolism dysregulation [8], and increased oxidative stress and neuroinflammation [9] have been proposed as potential causal factors. In line with that, for several years, the possibility of an interaction between the brain and skeletal muscle has been introduced [10]. Also, the correlation between cognitive and muscle function has been suggested by a recent human study [11]. Among possible mediators of the connection such as myokines, metabolites, muscular enzymes, and hormones, myokines have been investigated as the most potential mediator which could influence the brain and cognitive functions [10,12,13,14].

Myokines are signaling peptides released from the contraction of skeletal muscle and mediate cellular communication under physical exercise conditions [15]. For several years, exercise has been demonstrated as an effective intervention for amelioration of cognitive function in obese conditions. Previous investigations have suggested resistant exercise as an effective type of exercise in sarcopenic obesity [16] since it improves body composition [17], muscle strength [18], and cognitive function in older adults with cognitive frailty [19]. One possible way that physical exercise improves cognitive function is the increase in the secretion of cognitive function-related myokines. Cognitive function-related myokines have been demonstrated to regulate the metabolism of not only skeletal muscle but also the brain. A recent review also concluded that molecular mechanisms underlying exercise modulating cognitive performance are expected to be due to up-regulated neurogenesis and synaptic plasticity mediated by myokines [20].

Exercise-induced improvement of cognitive function has been suggested to be mediated by myokines, such as brain-derived neurotrophic factor (BDNF), cathepsin B (CTSB), irisin, insulin-like growth factor I (IGF-1), and interleukin-6 (IL-6) [20]. BDNF is a secretory neurotrophic factor that is produced in both brain and skeletal muscle, mainly regulated by physical exercise. BDNF contributes to the enhancement of cognitive function by encouraging neuronal survival, growth, and differentiation. Recently, it has been suggested that a reduction in BDNF is related to an obese state (although it is not clear [21]) and neuronal atrophy in the aging brain [22]. Irisin is a peroxisome proliferator-activated receptor-gamma coactivator 1-alpha (PGC-1α)-dependent myokine, a secreted form of fibronectin type III domain-containing protein 5 (FNDC5). Exercise is known to up-regulate gene expression of FNDC5 in the liver via the stimulation of PGC-1α expression level in the skeletal muscle [23]. Moreover, IGF-1 and CTSB are produced during exercise from the skeletal muscle. A recent randomized controlled trial (RCT) has demonstrated that in sarcopenic obesity, resistant exercise elevated the serum level of IGF-1 [24]. It has been demonstrated that irisin and CTSB mediate the positive effects of physical exercise on the brain via stimulation of BDNF synthesis with activation of protein kinase B (Akt) and the extracellular signal-regulated kinases 1/2 (ERK1/2) signaling pathway [14]. Lastly, IL-6 is a myokine that is secreted up to 100-fold in plasma as a response to muscle contraction. It has been suggested that exercise-induced IL-6 in the brain could be neuroprotective since the inflammation level was decreased [25]. Thus, it can be deduced that exercise-induced BDNF, irisin, IGF-1, and CTSB travel cross the blood-brain barrier (BBB) and act as a mediator of the exercise-induced improvement of cognitive function in sarcopenic obesity.

Examination of the molecular mechanisms related to muscle and brain interaction in type 2 diabetes mellitus (T2DM) showed that T2DM status led to aberrant energy metabolism in skeletal muscle and brain via dysregulation of myokines and hepatokines [26]. According to this study, it can be deduced that myokines could potentially have critical roles in central metabolic functions and possibly mediate cognitive function in sarcopenic obesity. Recent RCT showed that exercise and disconnected sitting improved working memory accompanied by elevated BDNF level in the aged population [27]. Even though several researchers suggested that modulation of myokines is an effective way to prevent the exacerbation of sarcopenia obesity [28,29], its underlying molecular mechanism is not fully elucidated. Moreover, changes and effects of myokines in both skeletal muscle and brain after resistant exercise have not been investigated. Thus, this study aimed to investigate the potential effects of resistant exercise on cognitive and muscle function in sarcopenic obese middle-aged mice, focused on cognitive function-related myokines in skeletal muscle, plasma, and hippocampus.

## 2. Materials and Methods

### 2.1. Sarcopenic Obesity Induction and Experimental Design

Eight-month-old C57BL/6J male mice (*n* = 30) from Janvier Labs (Rte du Genest, France0) were acclimated for 1 week under consistent conditions (temp: 22 ± 1 °C, humidity: 50 ± 5%, 12-h light-dark cycle). For sarcopenic obesity induction, mice were randomly divided into two groups: a normal diet-fed group (CON; *n* = 10) and a high-fat diet-fed group (HF; *n* = 20). The HF group was fed a 60% kcal high-fat diet (D12492; Research Diets, New Brunswick, NJ, USA), and the ND group was fed a 10% kcal fat control diet (D12450J; matching sucrose to D12492, Research Diets, New Brunswick, NJ, USA) for 8 weeks [30,31], provided ad libitum. After sarcopenic obesity induction, the HF group was randomly divided into two groups: non-exercise high-fat diet-fed group (OB; *n* = 10) and exercise high-fat diet-fed group (EX; *n* = 10). An additional 8 weeks of resistant exercise was implemented in the EX group shown in Figure 1. As a resistant exercise, ladder climbing was performed with a homemade 1 m ladder (1.5 cm grid steps, inclined 85~90°). Specifically, the exercise was performed 5 times weekly in the morning and gradually increased the load up to 20% of B.W. with a minimum of 8 to a maximum of 12 repetitions shown in Figure 2. Intensification of the exercise was implemented by adding weight to the animal’s tail and increasing repetitions [32]. Mice were motivated to climb the ladder by a gentle and soft pat on their body [33]. All of the experimental protocols related to animals were approved by the Institutional Animal Care and Use Committee of Kyung Hee University [KHSASP-21-247].

### 2.2. Muscle Function Test

For assessment of the physical function of skeletal muscle, the grip strength test was performed using a grip strength meter (Grip test package GS3 (25 N); Harvard Apparatus, US). The mice were directed to grasp the sensor bar, and the tail of the mice was gently pulled horizontally with the body at a fixed speed until their forelimbs were released [34]. The highest level of forelimb grip force was recorded, and the mean value of six repetitions was used.

### 2.3. Cognitive Function Test

Cognitive function was measured using a Y-shaped maze (three arms intersecting at 120°, 45 cm length × 35 cm height × 10 cm width). Each mouse was placed at the intersection of three arms and freely moved for 8 min. Entry into an arm was counted when the mouse’s end of the tail was inside an arm [35]. The percentage of spontaneous alternations was calculated using the following formula: [(number of spontaneous alternations)/(total number of entries − 2)] × 100.

### 2.4. Body Composition Analysis

Body composition was calculated by dual-energy X-ray absorptiometry (Medicors, InAlyzer, Gyeonggi-do, Korea) before and after exercise intervention. Each mouse was placed on the scanner bed after ketamine and xylazine anesthesia. The percentages of both fat mass and lead mass were used.

### 2.5. Histological Assays

The brain (hippocampus) and muscle (quadricep) were fixed with 10% formalin solution and were embedded in paraffin. The tissues were cut into 4 µm slides and stained with hematoxylin and eosin (H&E). The stained sections of tissues were observed using an optical microscope (Nikon ECLIPSE Ci, Tokyo, Japan). The mean density of neurons was calculated by quantifying the dentate gyrus area [36] using ImageJ software (NIH, Bethesda, MD, USA).

### 2.6. Western Blot Analysis

The brain (hippocampus) and skeletal muscle (quadriceps) were homogenized in the lysis buffer (10 mM HEPES, 10 mM KCl, 0.5 mM DTT, 1.5 mM MgCl2, 0.05% NP40) with protease inhibitor (Thermofisher, MA, USA). The homogenates were incubated for 1 hour at 4 °C and centrifuged at 845× *g* for 15 min at 4 °C. The collected supernatants were re-centrifuged at 18,407 g for 30 min at 4 °C, then the final supernatants were used for cytosol protein analysis. The concentration of protein was measured by Bicinchoninic acid (BCA) (Thermofisher, MA, USA) methods. The antibodies used in this study are; brain-derived neurotrophic factor(BDNF), peroxisome proliferator-activated receptor-gamma coactivator 1-alpha (PGC-1α), protein kinase B (pAkt, Akt), tropomyosin receptor kinase B (TrkB), interleukin-6 (IL-6), tumor necrosis factor-alpha (TNF-α), extracellular signal-regulated kinases 1/2 (pERK, ERK) (Santa Cruz Biotechnology, CA, USA, 1:200), 4-Hydroxynonena (4HNE) (BD Biosciences, NJ, USA, 1:500), C-Reactive Protein (CRP), irisin, insulin-like growth factor 1 (IGF-1) (Abcam, Cambridge, MA, USA, 1:1000), and α-tubulin (Sigma-Aldrich, St. Louis, MO, USA, 1:4000). Protein signals were detected using the ECL luminol reagent (Biorad, Hercules, CA, USA) and quantified with the Syngene GeneSanp (Syngene, Cambridge, UK, USA).

### 2.7. Statistical Analysis

All the numerical data were displayed as mean ± standard deviation (SD) drawn using an SPSS software (SPSS Inc., Chicago, IL, USA). A significant difference was analyzed by one-way ANOVA according to Duncan’s multiple range test. A significance level of *p* < 0.05 was implemented. Pearson’s correlation coefficient was implemented to evaluate the correlation between muscle and brain function.

## 3. Results

### 3.1. Induction of Sarcopenic Obesity

After 8 weeks of a high-fat diet, the OB and EX groups had significantly higher fat mass and body weight and lower lean mass compared to the CON group. Eight weeks of resistant exercise did not significantly change the body composition but suppressed the body weight increase of animals shown in Table 1.

### 3.2. Cognitive and Muscle Function and Their Correlation

After 8 weeks of resistant exercise, changes in cognitive and muscle function in sarcopenic obesity were measured. The OB group had significantly decreased cognitive and muscle function compared to the CON group. Resistant exercise significantly increased cognitive and muscle function in a sarcopenic obese state. Cognitive function demonstrated by % of alteration in Y maze is positively correlated with grip strength in all mice used in the experiment, as shown in Figure 3.

### 3.3. Morphological Analysis of Hippocampus

Morphological changes after 8 weeks of resistant exercise were investigated in the hippocampus. The OB group had a significantly lower density of neurons in the hippocampus compared to the CON group. In a sarcopenic obese state, resistant exercise significantly increased the density of neurons in the hippocampus. The OB groups had more severe apoptosis, vacuolation, and degradation of neuronal cells [37] compared to the CON and the RE groups shown in Figure 4.

### 3.4. Cognitive Function-Related Myokines and Energy Metabolism in Skeletal Muscle

The OB group had significantly lower protein levels of BDNF, tropomyosin receptor kinase B (Trk-B), irisin, and CTSB, but not in PGC-1α. Resistant exercise significantly up-regulated the protein levels of BDNF, PGC-1α, irisin, and CTSB. There were no significant differences in IL-6 and IGF-1 among the groups shown in Figure 5a.

In the skeletal muscle, the protein level of pAkt in the OB group was significantly lower than that of the CON group. Resistant exercise significantly increased the ratio of the pERK/ERK in sarcopenic obesity. There were no significant differences in Akt, pAkt/Akt, ERK, and pERK among the groups, as shown in Figure 5b.

### 3.5. Cognitive Function-Related Myokines, Energy Metabolism, and Oxidative stress and Inflammation in Hippocampus

The OB group had significantly lower protein levels of BDNF and CTSB compared to the CON group, but not in PGC-1α and irisin. Resistant exercise significantly up-regulated the protein levels of BDNF, PGC-1α, and CTSB. There were no significant differences in Trk-B, IL-6, and IGF-1 among the groups, as shown in Figure 6a.

In the hippocampus, the protein level of Akt in the OB group was significantly higher than that of the CON group, not in pAkt and pAkt/Akt. Resistant exercise significantly increased the ratio of the pAkt/Akt in sarcopenic obesity. There were no significant differences in ERK, pERK, and pERK/ERK among the groups, as shown in Figure 6b.

Compared to the CON group, the OB group had a significantly higher protein level of C-reactive protein (CRP) in the hippocampus. Resistant exercise significantly down-regulated the protein level of CRP in sarcopenic obese conditions. There were no significant differences in 4-Hydroxynonenal (4HNE) and tumor necrosis factor-α (TNF-α) among the groups, as shown in Figure 6c.

### 3.6. Cognitive Function-Related Myokines in Plasma

No significant differences were found between the CON and the OB group in the protein level of PGC-1α and BDNF. Resistant exercise significantly up-regulated the protein level of PGC-1α and BDNF. There were no significant differences in irisin, CTSB, and IL-6 among the groups, as shown in Figure 7.

## 4. Discussion

In the last decade, it has been demonstrated that skeletal muscle produces and secretes various kinds of myokines that induce either autocrine, paracrine, or endocrine effects on other metabolic organs via plasma. Many studies have examined myokines as potential biomarkers that reflect whole-body metabolism homeostasis [28]. As myokines are involved in the regulation of energy metabolism homeostasis of key organs [38], diverse pathological conditions such as obesity, sarcopenia, and senescence influence the level of myokine secretion [39,40]. Moreover, a recent review concluded that myokines can potentially act as potential diagnostic biomarkers in sarcopenic obesity [28]. This research found that in a sarcopenic obese state, the level of several myokines in the muscle (BDNF, irisin, and CTSB) and the hippocampus (BDNF, CTSB) significantly were lower than those in the normal state. It can be inferred that myokine varies reactively depending on the biological state and could be a reliable diagnostic biomarker of different pathological conditions. In addition, in this study, the reduction of hippocampal BDNF and CTSB might contributed to the cognitive disability in the sarcopenic obese state [41].

It is very well-known that physical exercise stimulates the production and secretion of myokines from the muscle, which influence the crosstalk among other organs [42]. A recent review suggested that myokines, produced in the skeletal muscle in response to exercise, play roles in the communication between the muscle and other organs including the brain. As myokines exert biological effects on cognition, myokines could be useful biomarkers for neurodegeneration [14]. Furthermore, a recent meta-analysis demonstrated that especially resistant exercise is pivotal in improving body composition and muscle strength in sarcopenic obesity [43].

The current study demonstrated that resistant exercise ameliorated cognitive and muscle functions in sarcopenic obesity. Additionally, this research demonstrated a positive correlation between cognitive and muscle function. Like this trend, BDNF, CTSB, and PGC-1α were simultaneously up-regulated in both the hippocampus and the skeletal muscle after resistant exercise. This implies that there is a possibility of a connection between the skeletal muscle and the brain. In addition, the muscle-derived cognitive function-related myokines (MCFM) could be the mediator of the connection, and have contributed to the cognitive improvement in sarcopenic obesity.

The hippocampus is the site of the brain where the process of learning and memory actively progresses. Physical exercise has been used as an effective neurotrophic stimulus, increasing BDNF level in the hippocampus [44]. This study demonstrated that resistant exercise selectively up-regulated MCFM (BDNF and CTSB) with PGC-1α in both the hippocampus and skeletal muscle in sarcopenic obesity. Consistent with our results, recent research showed that endurance exercise up-regulated BDNF expression through PGC-1α and irisin in the hippocampus of mice [45]. In addition, the previous investigation demonstrated that physical exercise reduced apoptosis signaling and enhanced survival pathways in the hippocampus of aged rats with increased expression of PGC-1α [46]. Moreover, previous research suggested that the potential connection between exercise and cognitive improvement is hippocampal neurogenesis [47]. BDNF has critical roles including cell proliferation, differentiation, and survival in the central nervous system neurons [48]. In line with alleviated histopathological changes in the hippocampus after resistant exercise, we expect that promoted neurogenesis, specifically neuronal proliferation and differentiation by resistant exercise contributed to the increased neuronal density and cognitive ability. Therefore, it can be deduced that resistant exercise-induced MCFM contributed to cognitive enhancement in sarcopenic obesity.

BDNF, which is produced in both muscle and the brain, promotes the development, survival, and functions of neurons [49] mediated by the activation of energy metabolism-related Akt and ERK signaling [50,51]. Previous research demonstrated that, in skeletal muscle, activation of ERK1/2 signaling inhibits muscle damage in muscle-damaged mice [52]. PGC-1α plays several pivotal roles in organs; lack of PGC-1α is associated with neurodegeneration [45], and over-expression of PGC-1α protects aging mice from muscle wasting [53]. A previous RCT showed that CTSB level is correlated with hippocampus-dependent memory function, indicating CTSB as a crucial mediator of exercise effects on cognition [54].

The whole body becomes metabolically unstable in a sarcopenic obese condition, inducing energy metabolism dysregulation in multi-organs including the skeletal muscle and hippocampus [55]. Collectively, this study suggested that resistant exercise promoted energy metabolism homeostasis: pAkt/Akt and PGC-1α in the hippocampus and pERK/ERK and PGC-1α in skeletal muscle. It is expected that elevated energy metabolism homeostasis contributed to both cognitive and muscle function enhancement in this study. Moreover, reinforced MCFM (BDNF and CTSB) secretion by energy metabolism homeostasis potentially triggered neurogenesis and inhibited neurodegeneration in the hippocampus, eventually strengthening cognitive function. Therefore, it can be inferred that resistant exercise alleviated the complications of sarcopenic obesity through an elevation of MCFM level in the interaction between skeletal muscle and the brain.

It is careful to say that MCFM reached the hippocampus via plasma and increased cognitive function by resistant exercise in sarcopenic obesity, in form of cross-talk. However, our results showed that the levels of BDNF and PGC-1α simultaneously increased in the skeletal muscle, plasma, and hippocampus. Along with that, the recovery of abnormal morphological changes in the hippocampus by resistant exercise supports the linkage between the skeletal muscle and the brain mediated by MCFM. Selective changes in plasma MCFM levels in the plasma could be explained by active responses to physical exercise by other secretomes, which also secrete myokines.

Previous studies demonstrated that obesity in aging is interconnected with hippocampal inflammation, which exacerbates blood-brain barrier (BBB) disruption, synaptic dysfunction, and cognitive impairment [56]. This study clearly illustrated that increased hippocampal inflammation (CRP) was significantly ameliorated by resistant exercise in sarcopenic obesity. This result is consistent with a previous study that showed resistance training improved cognitive performance and reduced neuroinflammation in the hippocampus of Alzheimer’s disease mice [57]. In addition, a previous clinical study demonstrated a negative correlation between CRP and cognitive function [58]. Furthermore, previous research demonstrated that inflammation extensively impaired hippocampal neurogenesis, which was detrimental to neuronal survival in rats. However, microglia activation can protect neurons by promoting the release of neurotrophic molecules [59]. It was concluded that the metabolic-inflammatory interplay of the brain is composed of interconnected energy metabolism, neuroinflammation, and redox regulation [60]. The current research suggested that increased MCFM (BDNF and CTSB) in the brain protected neurons from degeneration by suppressing inflammation and fostering energy metabolism, in line with morphological analysis in the hippocampus. Therefore, it can be deduced that proliferated neurons assisted by MCFM contributed to improvement in cognitive function.

One of the consequences of aging skeletal muscle is the impaired function of IGF-1-producing macrophages and an up-regulated concentration of pro-inflammatory cytokines [61]. A previous study demonstrated that the plasma levels of BDNF and irisin were not significantly different between aged-sedentary and aged-exercise rats [62]. Though our results were slightly different from a previous study that demonstrated up-regulation of hippocampal irisin by endurance exercise [45], it can be expected that adverse physiological changes in aging progress disrupted MCFM secretion in sarcopenic obesity. Furthermore, there is a possibility that the resistant exercise we implemented was not enough to bring a significant difference in the secretion of MCFM. Recent studies concluded that, due to inconsistency, understanding the mechanisms of myokine is a fundamental factor in counteracting aging-related dysfunction, at least in the skeletal muscle [40,63].

BDNF and its high-affinity receptor Trk-B are widely distributed in regions of the brain. Specifically, BDNF/Trk-B is highly expressed in the dentate gyrus of the hippocampus and plays a critical role in neurophysiological responses [64,65]. Although hippocampal Trk B was not elevated unlike BDNF in this study [66], a much higher level of BDNF might contribute to cognitive function enhancement by a higher binding chance with Trk-B.

## 5. Conclusions

This study demonstrated that resistant exercise significantly normalized the dysfunction in both muscle and the brain in a sarcopenic obese state summarized in Figure 8. Based on a positive correlation between muscle and brain function, we investigated the effects of resistant exercise on sarcopenic obesity focusing on muscle-derived cognitive function-related myokines (MCFM) with energy metabolism. Resistance exercise increased MCFM (BDNF and CTSB) level of skeletal muscle and hippocampus and energy metabolism homeostasis (Akt and ERK) in a sarcopenic obese state, with clear alteration shown in BDNF level. Results show the potential linkage between the skeletal muscle and hippocampus and imply a positive effect of resistant exercise on cognitive and muscle function in sarcopenic obesity. Moreover, it is expected that decreased hippocampal inflammation assisted exercise-induced cognitive improvement in sarcopenic obese conditions. Thus, modulation of MCFM via resistant exercise could be a possible way of relieving cognitive and muscle dysfunction for both sarcopenia and obesity patients

## Figures and Tables

**Figure 1 biomedicines-10-02529-f001:**
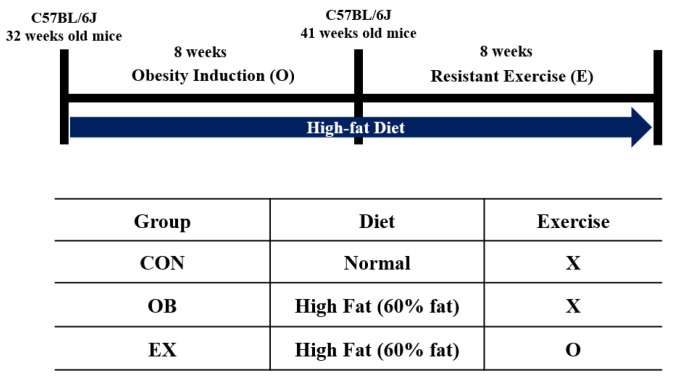
Experimental design. X: non-exercise, O: exercise.

**Figure 2 biomedicines-10-02529-f002:**
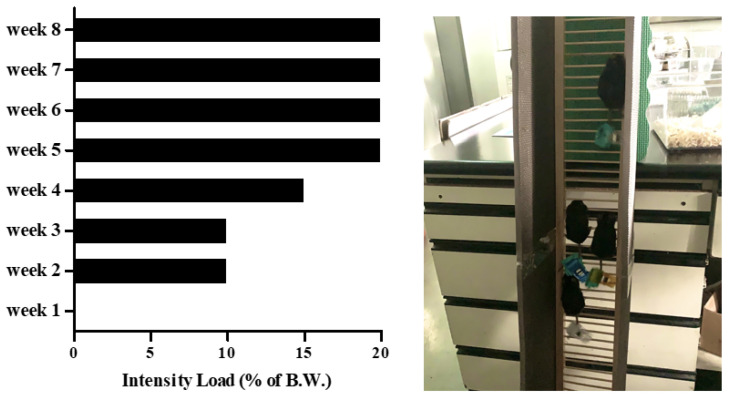
The incremental intensity load applied to the resistant exercise.

**Figure 3 biomedicines-10-02529-f003:**
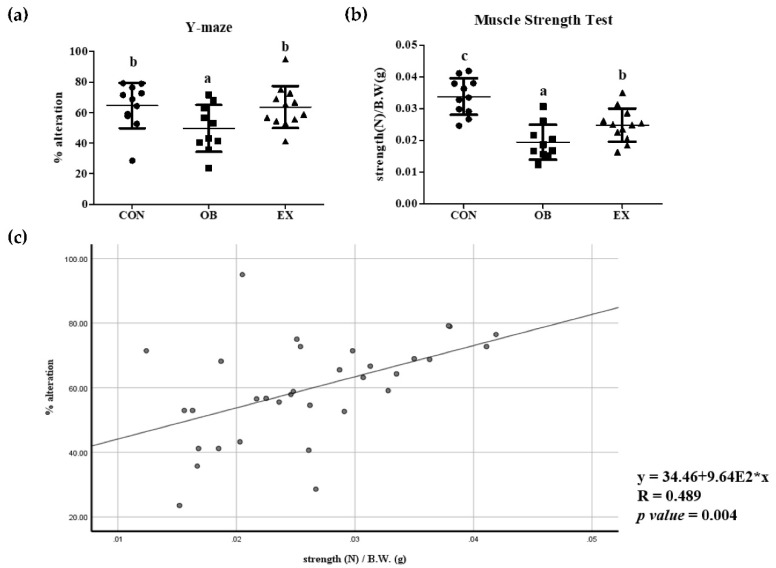
Effects of resistant exercise on (**a**) cognitive and (**b**) muscle function, and (**c**) correlation between cognitive and muscle functions. Values are means ± SD (*n* = 10~12). Mean values with the same superscript letter (a, b and c) are not significantly different (*p* < 0.05). *: multiply.

**Figure 4 biomedicines-10-02529-f004:**
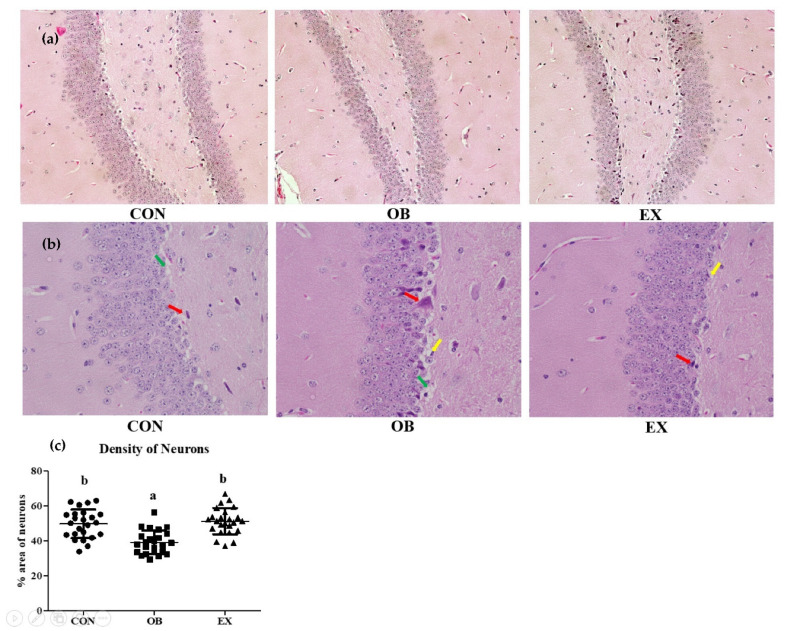
Effects of resistant exercise on the hippocampus (dentate gyrus) morphology in sarcopenic obese mice. (**a**) hippocampus (X200), (**b**) hippocampus (X400), and (**c**) density of neurons. Six random sites were examined. The red arrows represented apoptosis, the green arrows represented vacuolation, and the yellow arrows represented the degradation of neuronal cells. Values are means ± SD (*n* = 3~4). Mean values with the same superscript letter (a and b) are not significantly different (*p* < 0.05).

**Figure 5 biomedicines-10-02529-f005:**
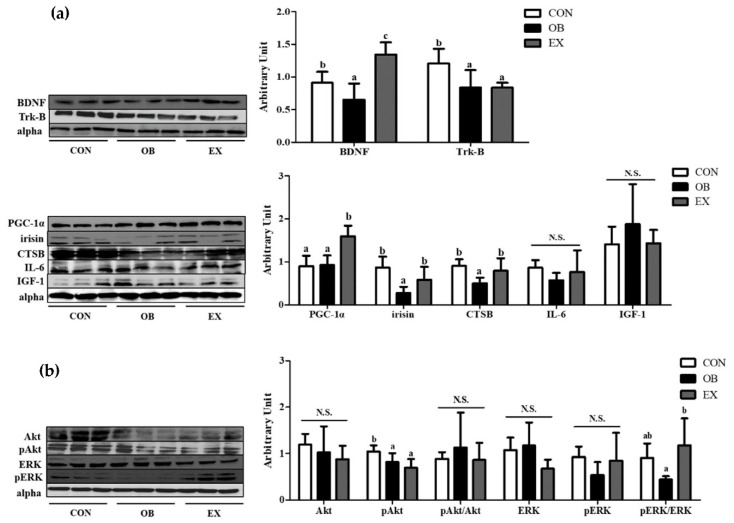
Effects of resistant exercise on skeletal muscle. (**a**) cognitive function-related myokines and (**b**) energy metabolism in sarcopenic obese mice. Values are means ± SD (*n* = 5~6). Mean values with the same superscript letter (a, b and c) are not significantly different (*p* < 0.05). BDNF: brain-derived neurotrophic factor, Trk-B: tropomyosin receptor kinase B, PGC-1α: peroxisome proliferator-activated receptor-gamma coactivator 1-alpha, CTSB: cathepsin B, IL-6: interleukin-6, IGF-1: insulin-like growth factor 1, Akt: protein kinase B, ERK1/2: extracellular signal-regulated kinases 1/2. N.S.: no significance.

**Figure 6 biomedicines-10-02529-f006:**
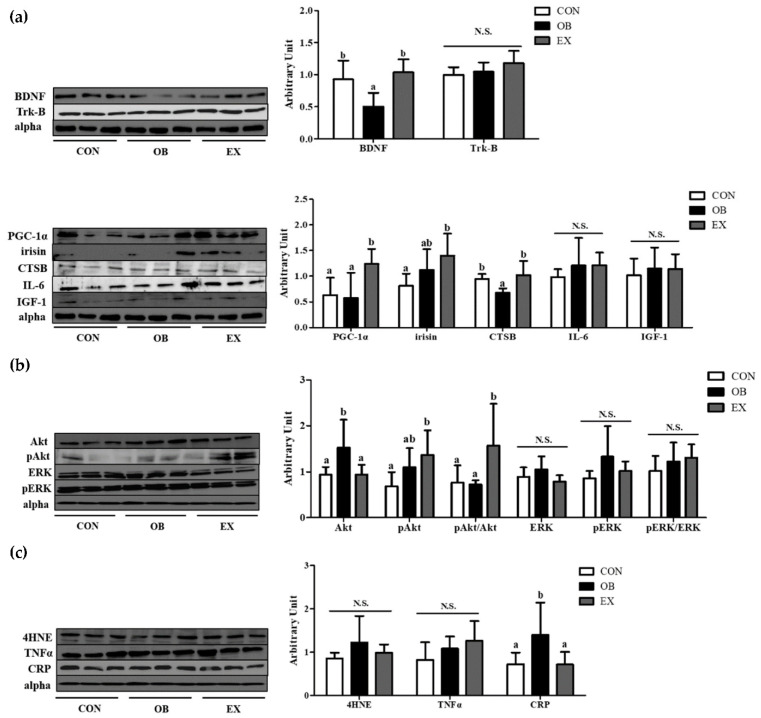
Effects of resistant exercise on hippocampal (**a**) cognitive function-related myokines and (**b**) energy metabolism and (**c**) oxidative stress and inflammation in sarcopenic obese mice. Values are means ± SEM (*n* = 5~6). Mean values with the same superscript letter (a and b) are not significantly different (*p* < 0.05). BDNF: brain-derived neurotrophic factor, Trk-B: tropomyosin receptor kinase B, PGC-1α: peroxisome proliferator-activated receptor-gamma coactivator 1-alpha, CTSB: cathepsin B, IL-6: interleukin-6, IGF-1: insulin-like growth factor 1, Akt: protein kinase B, ERK1/2: extracellular signal-regulated kinases 1/2, 4HNE: 4-Hydroxynonenal, TNF-α: tumor necrosis factor-alpha, CRP: C-Reactive Protein. N.S.: no significance.

**Figure 7 biomedicines-10-02529-f007:**
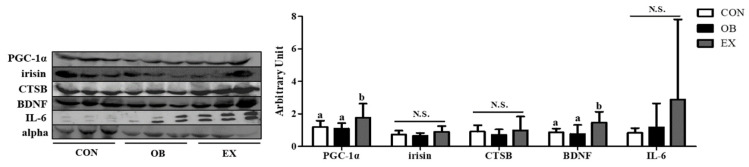
Effects of resistant exercise on plasma cognitive function-related myokines in sarcopenic obese mice. Values are means ± SD (*n* = 5~6). Mean values with the same superscript letter (a and b) are not significantly different (*p* < 0.05). BDNF: brain-derived neurotrophic factor, PGC-1α: peroxisome proliferator-activated receptor-gamma coactivator 1-alpha, CTSB: cathepsin B, IL-6: interleukin-6. N.S.: no significance.

**Figure 8 biomedicines-10-02529-f008:**
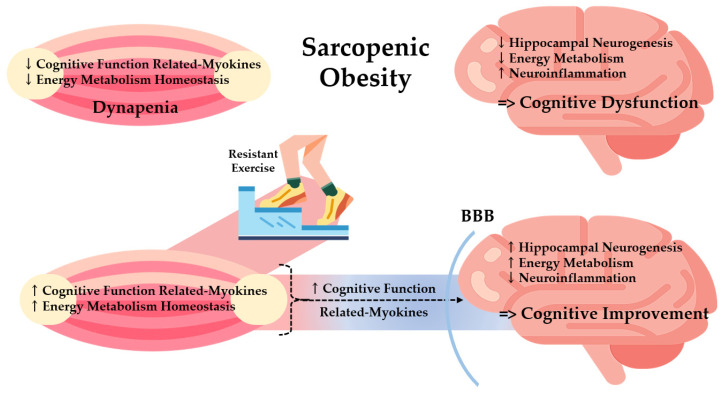
Potential effects of resistant exercise on cognitive and muscle function mediated by myokines in sarcopenic obesity. BBB: Brain Blood Barrier.

**Table 1 biomedicines-10-02529-t001:** Body composition and body weight of sarcopenic obesity-induced animals.

Group		CON	OB	EX
Body Weight (g)				
Before Exercise		31.33 ± 2.39 ^a^	39.39 ± 3.93 ^b^	40.14 ± 4.63 ^b^
After Exercise		31.70 ± 2.63 ^a^	43.64 ± 5.27 ^b^	42.01 ± 5.41 ^b^
Difference		0.38 ± 1.08 ^a^	4.25 ± 2.03 ^c^	1.87 ± 1.63 ^b^
Body Composition (% of Total Mass)				
Before Exercise	Fat Mass	22.54 ± 2.94 ^a^	35.26 ± 6.28 ^b^	36.58 ± 5.41 ^b^
	Lean Mass	74.93 ± 2.87 ^b^	62.60 ± 6.12 ^a^	61.35 ± 5.31 ^a^
After Exercise	Fat Mass	24.77 ± 3.55 ^a^	38.34 ± 7.43 ^b^	38.90 ± 5.54 ^b^
	Lean Mass	72.58 ± 3.47 ^b^	59.53 ± 7.24 ^a^	58.91 ± 5.45 ^a^
Difference	Fat Mass	2.26 ± 2.50	3.09 ± 2.80	2.32 ± 1.76
	Lean Mass	−2.35 ± 2.46	−3.08 ± 2.71	−2.45 ± 1.77

Values are means ± SD (*n* = 10~12). Mean values with the same superscript letter (a and b) are not significantly different (*p* < 0.05).

## Data Availability

Not applicable.

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
