# Peer review of "Potential Effects of Resistant Exercise on Cognitive and Muscle Functions Mediated by Myokines in Sarcopenic Obese Mice"

_biomedicines, 2022, doi:10.3390/biomedicines10102529_

Round 1
Reviewer 1 Report
In this study the Authors focused on the investigation of the effects of resistant exercise on cognitive and muscle function in sarcopenic obese middle- aged mice. Based on a positive correlation between muscle and brain function, they examined the effects of resistant exercise on sarcopenic obesity focusing on muscle-derived cognitive function-related myokines with energy metabolism. From the obtained results, Resistance exercise increased BDNF and CTSB level in skeletal muscle and hippocampus and energy metabolism homeostasis (Akt and ERK) in a sarcopenic obese state.
This is a very interesting, important and innovative study
There are several recommendations:
1 Abstract.
Please, remove parentheses where it is possible.
2 Methods.
Please indicate whether the mice exercised in the morning, afternoon, or evening.
3 Methods
Please, extend legend to Figure 1.
4. Results
Figure 3 A and B
Please, provide plot with values for separate points
Figure 4 A
If it is possible, please, provide picture of better quality or of higher magnification (x400 would be quite good)
Figure 4 B
Please, provide plot with values for separate points
5. There is one important question:
What is the Nature of hippocampus neuron density recovery after exercises?
Cell hypertrophy or, perhaps cell polyploidization, or proliferation ?
One more source of adult neurogenesis - is neuron formation from proliferating neuronal precursor cells…
How do you think, which particular way of neurogenesis resistant exercises can stimulate?
Please discuss interesting question ….
If it is possible, it also would be good to evaluate and characterize the effects of resistant exercises on neuron morphology in more detail…
It would be good to insert several phrases in Result section describing cell morphology or morphometry, including, perhaps cell area, features of DNA instability and apoptosis,
Density of staining…. Or… cell number per unit area (i.e. per one microscopic field at magnification 400)
You can do this easily just on the histological sections that you already have. Sure it is quite easy foryou. Please, do this finishing step… Your beautiful story should be well supported by the excellent image analysis.

Reviewer 2 Report
This work discussed the potential effects of resistant exercise on cognitive and muscle function mediated by myokines in sarcopenic obese mice. There are some issues in this manuscript that should be addressed as follows:
Title: The word “function” should be replaced with “functions”.
Abstract:
- The meaning of the abbreviations should be clearly addressed at their first mention, e.g. BDNF, 20 CTSB, PGC-1a.
- The abstract should end with a concluding statement.
· Introduction:
1. The novel aspects in this study should be clarified as there are previous reports that discussed a closely similar topic.
2. The last sentence in the first paragraph “Reduction in cognitive and muscle functions seriously hinders an individual’s independence and quality of life. In this regard, investigating how to mitigate declined cognitive and muscle functions in sarcopenic obesity is important” in the “Introduction” needs a reference.
· Materials and methods:
1. Page 3 Line 101: The word “was” should be replaced with “were”.
2. A reference for the sarcopenic obesity induction should be provided.
3. A reference for the muscle function test and the cognitive function test should be added.
4. The method of quantification of the mean density of neurons and myofiber area should be mentioned.
5. The exact source, concentrations and the catalogue numbers of the used kits and chemicals should be mentioned.
6. How did you know that the animals were acclimatized?
7. Page 4 Lines 168, 169, 170: The meaning of the abbreviations should be clearly addressed at their first mention.
8. I think that it is preferrable to express the results as mean ± SD.
· Results:
1. In figure 4A, arrows should be added to show the positive histopathologic findings.
2. In all figures and table, the meaning of the symbols (a, b, ….etc) that identify the significant differences between the different groups should be added.
· Discussion:
1. Page 8: The last sentence in the first paragraph of the “Discussion” needs a reference.
2. The discussion should provide more details to analyze of the results of the different parameters measured in the present study.
· Conclusion: The clinical value of the findings of this study should be mentioned in the “Conclusion” section.
· General comments:
1. The manuscript should be revised by English-naïve speaker to improve the quality of the language.
2. The manuscript should be checked regarding the grammatical errors and plagiarism.
